A novel non-invasive nociceptive monitoring approach fit for intracerebral surgery: a retrospective analysis

Ruemmler Robert robert.ruemmler@email.de
Moravenova Veselina
Al-Butmeh Sandy
Fukui-Dunkel Kimiko
http://orcid.org/0000-0002-9959-3189 Griemert Eva-Verena eva-verena.griemert@unimedizin-mainz.de
Ziebart Alexander
Department of Anesthesiology, University Medical Center Mainz , Mainz , Germany
Abdullah Jafri
Electronic publication date: 2024 Jan 16
Publication date: 2024
Volume: 12
Electronic Location ID: e16787
Received 2023 Aug 17; Accepted 2023 Dec 20
Copyright: © 2024 Ruemmler et al.
Copyright year: 2024
Copyright holder: Ruemmler et al.
License: This is an open access article distributed under the terms of the Creative Commons Attribution License, which permits unrestricted use, distribution, reproduction and adaptation in any medium and for any purpose provided that it is properly attributed. For attribution, the original author(s), title, publication source (PeerJ) and either DOI or URL of the article must be cited.
License URL: https://creativecommons.org/licenses/by/4.0/

Keywords: Nociception level, Neurosurgery, Anesthesia, Monitoring, Hypoanalgesia, Hyperanalgesia

Funding: The authors received no funding for this work.

==============================
Background

Measuring depth of anesthesia during intracerebral surgery is an important task to guarantee patient safety, especially while the patient is fixated in a Mayfield-clamp. Processed electro-encephalography measurements have been established to monitor deep sedation. However, visualizing nociception has not been possible until recently and has not been evaluated for the neurosurgical setting. In this single-center, retrospective observational analysis, we routinely collected the nociceptive data via a nociception level monitor (NOL®) of 40 patients undergoing intracerebral tumor resection and aimed to determine if this monitoring technique is feasible and delivers relevant values to potentially base therapeutic decisions on.

Methods

Forty patients (age 56 ± 18 years) received total intravenous anesthesia and were non-invasively connected to the NOL® via a finger clip as well as a bispectral-index monitoring (BIS®) to confirm deep sedation. The measured nociception levels were retrospectively evaluated at specific time points of nociceptive stress (intubation, Mayfield-positioning, incision, extubation) and compared to standard vital signs.

Results

Nociceptive measurements were successfully performed in 35 patients. The largest increase in nociceptive stimulation occurred during intubation (NOL® 40 ± 16) followed by Mayfield positioning (NOL® 39 ± 16) and incision (NOL® 26 ± 12). Correlation with BIS measurements confirmed a sufficiently deep sedation during all analyzed time points (BIS 45 ± 13). Overall, patients showed an intraoperative NOL® score of 10 or less in 56% of total intervention time.

Conclusions

Nociceptive monitoring using the NOL® system during intracerebral surgery is feasible and might yield helpful information to support therapeutic decisions. This could help to reduce hyperanalgesia, facilitating shorter emergence periods and less postoperative complications. Prospective clinical studies are needed to further examine the potential benefits of this monitoring approach in a neurosurgical context.

Trial registration

German trial registry, registration number DRKS00029120.

Background

The adequate assessment of sufficient depth of anesthesia during surgical procedures is paramount to prevent episodes of awareness and involuntary movement potentially leading to intraoperative injuries or long-term effects including post-traumatic stress syndrome (Lewis et al., 2019). Intracranial surgery in particular poses a specific challenge, since even the smallest movements can lead to intracerebral hemorrhage or severe spinal injuries due to the rigid fixation of the cranium inside a Mayfield clamp (Caird & Bolger, 2005). Additionally, patients with intracranial pathologies should be available for postoperative neurological assessment as soon as possible in order to recognize complications and react accordingly (Sarwal, 2021). However, hyperanalgesia with opioid overdosing or unnecessarily deep hypnosis and muscle relaxation can be detrimental to that goal.

Apart from clinical evaluation of the patient based on blood pressure, heart rate and breathing patterns, technical solutions have been developed over the years, allowing for assessment of deep hypnosis using processed electroencephalographic signals like the bispectral index (BIS) (Lewis et al., 2019). Unfortunately, objective, specific and reliable methods to determine nociceptive stimulation and adequate analgesia were not available for a long time. In recent years, several non-invasive, multimodal systems have been developed and aim to close that knowledge and quantification gap (Koschmieder et al., 2023; Nitzschke, Fischer & Funcke, 2021). The PMD-200 Nociception Level monitor (NOL®) is one such device, established by attaching a sensor to the patient’s finger. The multi-channel assessment simultaneously measures temperature, perfusion, movement and skin conductivity and processes those measurements to a dimensionless score reaching from 0 (no activity) to 100 (maximum activity/pain) with a threshold of sufficient analgesia from 10–35 (Sabourdin et al., 2022).

While there have been some pilot trials to validate the NOL® system for analgesic management (Sabourdin et al., 2022; Renaud-Roy et al., 2022), no assessment for intracranial procedures has been published.

The aim of this retrospective analysis was to test the feasibility and validity of the NOL® system in a standardized setting during intracranial surgery in order to enable future planning of prospective trials and potentially prevent hyperanalgesia and unnecessarily prolonged anesthesia emergence. We specifically focussed on defined perioperative situations with high nociceptive potential.

Methods

The data of 40 adult patients with American Society of Anesthesiologists physical status (ASA) 1–3 undergoing elective intracranial surgical procedures (tumor resections in supine position performed between January and November 2022) were retrospectively analyzed (study registration no. DRKS00029120, registration date 07-01-2022, ethics committee approval no. 2021-16201, regional ethics committee of Rhineland Palatine, Mainz, chairman Dr. Stephan Letzel). No specific consent was required. No exclusion criteria were defined. Selected patients received standardised anesthesia induction with remifentanil (0.2 µg kg−1 min−1) and propofol (2–3 mg kg−1). All patients received standardised hemodynamic monitoring (invasive blood pressure measurements, heart rate, oxygen saturation), as well as sedation monitoring via the bispectral-index monitoring system (BIS monitoring system, Medtronic, Minneapolis, USA). The PMD-200 Nociception Level (NOL®) measurement device (PMD-200; Medasense Biometrics Ltd., Ramat Gan, Israel) was established and adequate signal quality was confirmed prior to the first drug injection.

After the induction, an intraarterial line was established on the contralateral arm of the NOL® sensor and anesthesia was maintained via continuous propofol (5–10 mg kg−1 h−1) and remifentanil (0.1–0.5 µg kg−1 min−1) drips. Doses were adapted following the clinical reasoning of the responsible anesthetist, independent of a fixed treatment protocol or predetermined NOL® thresholds. All measurements were recorded continuously. At five distinct time points (baseline prior to anesthesia, intubation, Mayfield clamp positioning, incision and extubation) NOL®, BIS®, mean arterial pressure (MAP) and drug infusion rates of propofol and remifentanil were marked in the anesthesia protocol and the NOL® software, respectively, and evaluated retrospectively. Baseline was defined as first stable measurements of the patients prior to induction. As NOL® and BIS® values, due to method latency, the highest value within 2 min of the stimulation was taken into consideration. In case of NOL® signal detection problems for more than 20% of the total intervention, patient data was excluded from the analysis.

Postoperative pain levels were assessed via numeric pain rating scale (NPRS) ranging from 0 (no pain) to 100 (maximum pain) in increments of 10. Patients were evaluated in the post anesthesia care unit (PACU) at arrival and every 30 min after that. No non-opioid regimen was determined, acetaminophen (1 g), ibuprofen (600 mg) and/or metamizole (1 g) were given as prescribed by the anesthetist. The only opioid used in the PACU was piritramide.

Statistical analysis

All statistical analyses were performed using GraphPad Prism 9 software (GraphPad Software, Boston, MA, USA). Since this was a pilot analysis, no statistical planning was performed prior to data collection, rendering the evaluations presented primarily descriptive. Repeated measurements were analyzed with 2-way ANOVA, time-point comparisons were analyzed using Mann-Whitney test. P-values < 0.05 were considered significant.

Results

A total of 40 patients treated between January and October of 2022 were included (age 56 ± 18, ASA 2.5 ± 0.55, 55% male) in the retrospective analysis. Resected tumors included glioblastoma (n = 18), meningioma (n = 18) and intracranial metastatic lesions (n = 4). Five patients had to be excluded after selection for problems with the NOL® monitoring (less than 80% adequate data collection). In the other 35 patients, NOL® measurements achieved continuous calculation of the NOL Index independent of surgical stimulation (Fig. 1). NOL showed significantly increased values during nociceptive stimulation compared to baseline measurements (baseline 20.6 ± 10.5; intubation 40.8 ± 16.4; P < 0.0001, Mayfield 38.8 ± 15.4; P < 0.0001, incision 26.4 ± 12.3; P = 0.03, extubation 55.4 ± 17.5; P < 0.0001, Fig. 2).

Figure 1 NOL examples.

Example of a complete NOL® measurement graph during a surgical procedure (A) and a small cutout (B) focussed on patient intubation time point (arrow). The latency of NOL® measurements is described with 30–60 s, which can be verified with the delayed increase in (B).

Figure 2 NOL® measurements at five predetermined time points.

Nociception levels show a highly significant increase during stimulation with an additional pronounciation at extubation. (*; **** = P-value < 0.05; 0.0001 compared to baseline, ##; #### = P-value < 0.01; 0.0001 compared to extubation).

BIS levels showed no significant increase during surgical stimulation, implying sufficient hypnosis throughout the intervention (baseline 94.7 ± 4, intubation 43.8 ± 15, Mayfield 48.5 ± 12, incision 45.6 ± 14, extubation 80.4 ± 7, Fig. 3).

Figure 3 Comparison of BIS® and NOL® levels including the respective thresholds for supposedly sufficient analgesia and sedation respectively.

The graph shows adequate sedation during the whole time anesthesia was maintained (BIS® goal 40–60). Nociception level measurements clearly show levels above the respective threshold, suggesting inadequate analgesia for the respective stimuli (NOL® goal <25).

Patients showed a NOL smaller than 10 over 56% ± 16% of the intervention time, 39% ± 15% ranged from 10–35 and 4.7% ± 3% was above 35% suggesting insufficient analgesia.

Mean arterial blood pressure (MAP), heart rate (HR) and drug doses were continuously recorded and are summarized in Table 1.

Table 1 Vital parameters during surgery.

Value	Baseline	Intubation	Mayfield	Incision	Extubation	
MEAN (SD)						
HR	73(11)	69(16)	59(12)	56(8)	84(16)	
(bpm)	
MAP	96(12)	84(19)	88(17)	76(14)	93(13)	
(mmHg)	
NOL	21(10)	41(16)***	39(15)***	26(12)*	55(18)***	
(-)	
BIS	95(4)	44(15)	49(12)	46(14)	80(7)	
(-)	
Norepinephrine	0	0.12(0.17)	0.16(0.18)	0.20(0.23)	0.02(0.06)	
(mg/h)	
Remifentanil	0	0.2(0)	0.22(0.06)	0.23(0.06)	0	
(µg*kg−1min−1)	
Propofol	0	178(63)	6.5(1.6)	7.0(1.6)	0	
(mg/mg*kg−1h−1)	
Note:

Cumulative trial data at the respective time points. SD = standard deviation, HR = heart rate, bpm = beats per minute, MAP = mean arterial pressure, NOL = nociception level index, BIS = bispectral index. * = P < 0.05; *** = P < 0.001 compared to baseline.

Postoperative pain level documentation could only be analyzed for 27 patients due to incomplete documentation or direct transport to an intensive care unit. Pain levels showed moderate postoperative pain (NPRS5: 10 ± 20.2; NPRS30: 13.2 ± 20.1; NPRS60: 4.1 ± 12.8; NPRS120: 1.6 ± 4.7) with low opioid use (3.3 mg ± 5 mg).

Discussion

This study examined—for the first time—the feasibility and general plausibility of the PMD 200 Nociception level device in a small patient population undergoing intracerebral surgical procedures in order to enable future interventional trial protocols based on NOL®. The analyzed measurements showed a good correlation with known painful stimuli while simultaneously confirming deep sedation during those episodes via BIS® monitoring. The data also show a substantial amount of time during those procedures, were nociception levels appeared to be very low. This could enable future trials with targeted analgesic treatment protocols, potentially avoiding those hyperanalgesic periods. This, in turn, could facilitate earlier emergence and improved postoperative patient evaluation while still maintaining intraoperative safety standards.

Nociception level monitoring has been a research focus for several years now and different devices and approaches have been developed. Originating from modified somatosensory-evoked potential analyses (Bublitz et al., 2022) and functional magnetic resonance imaging (Pujol et al., 2023), systems were first developed to analyse direct sympathomimetic reactions like pupillary dilation reflex to assess intraoperative pain (Kim et al., 2022). In recent years, other, more easily applicable devices with multi-channel measurements and patented processing algorithms like the analgesia nociception index (ANI®) (Jean et al., 2022; Baroni et al., 2022) or the NOL® have been established for perioperative use (Edry et al., 2016; Stöckle et al., 2018; Martini et al., 2015), but prospective, randomised trials are sparce and usually specifically tailored for distinct surgical procedures (Funcke et al., 2021; Funcke et al., 2020). However, promising data could be shown regarding adequate postoperative pain prediction (Morisson et al., 2023; Koschmieder et al., 2023), detection of analgesia (Ghiyasinasab et al., 2022; Niebhagen et al., 2022) and reduction of opioid administration (Espitalier et al., 2021; Michalot et al., 2022; Sabourdin et al., 2022). Additionally, pain reduction after targeted opioid use (Fuica et al., 2023) and earlier emergence after NOL®-directed therapy (Renaud-Roy et al., 2022) have been reported. Unfortunately, all published studies to this date are very small, usually single-center and often not prospectively randomised.

The presented analysis supports the growing knowledge about potential benefits of NOL®-guided analgesia and offers basic data for prospectively designed intervention protocols. However, some limitations of the monitoring method have to be considered. According to clinical experience and the manufacturer, heart rate variability is a crucial factor for the underlying algorithm to properly assess all data collected by the device. In the clinical setting, this leads to problems with patients suffering from arrhythmias or under beta-blocker therapy as well as patients with compromised extremity perfusion or low blood pressure. Additionally, in analogy to the BIS® monitor, there is a latency between 30 to 90 s between pain stimuli and adequate NOL® signalling. This has to be accounted for when targeting therapy suggestions. While especially the use of beta-blockers has been shown to marginally influence results (Bergeron et al., 2022), additional factors like difficult mounting of the sensor due to high body mass index or extreme differences in patients’ heights have to be considered and might lead to problems with data collection. Another potential confounding factor of the comparison of NOL® data in this analysis specifically might stem from the decision, not to apply any exclusion criteria. This can result in the inclusion of patients with chronic pain medication who are used to opioids and require significantly higher analgetic doses. While there were few patients with opioids present in their previous medication plans, no significant influence on the overall data could be detected. Nevertheless, future trials would need to account for those factors in advance. Similarly, a more rigid and standardized postoperative pain management would need to be implemented, also defining eligible non opioids and supportive drugs.

The decision to set the baseline while the patients were still awake could also be controversial. While nociception monitoring has been shown to produce somewhat plausible values in awake patients undergoing surgery (Baroni et al., 2022) or physical therapy (Santella et al., 2022), movement being one of the analyzed parameters might cause substantial confounders when compared to sedated time points in the same patients. However, it can be expected, that NOL® levels at baseline in the already sedated patient would be even lower and the increases would only be more pronounced. The same argument naturally counts for the extubation, where patients will inadvertently move and potentially influence the NOL® values stronger, than the actual nociception might represent in reality. We also see this in our data with a distinct increase of the measurements at extubation time points. However, postoperative analgesic demand did not reflect any correlation suggesting actual hypoanalgesia. This only further supports the hypothesis that nociceptive measurements are not reliable in awake or emerging patient collectives.

Generally, it has to be stated that current nociceptive level measurement systems are never able to directly determine nociceptive activity. All of them rely on surrogate parameters, mostly derived from activity levels of the vegetative system. To increase the potential specificity of the provided information, the combination of existing systems could be attempted. However, we were only able to use the NOL®-system in our hospital for this study and the added benefit of another simultaneously used system could be discussed. At this point, no helpful analysis towards this topic is available,

While there have been studies suggesting an advantage of NOL®-guided analgesia regarding postoperative pain prediction and opioid use compared to standard clinical decision making (Morisson et al., 2023; Fuica et al., 2023), no large trials have been conducted and other studies could not show any benefits (Koschmieder et al., 2023). In our analysis, NOL® increase did not coincide with higher BIS® values, however, heart rate and MAP especially showed a distinct increase, suggesting clear clinical signs for potentially insufficient analgesia. This would have most likely been treated in any clinical context, independent of specific monitoring.

In our study, measurements in roughly 85% of the patients were successful, but intermittent signal loss can occur and has to be accounted for. The data sets that had to be excluded in our study were mainly caused by the aforementioned factors, i.e., substantial arrhythmias, very small hands or extremely adipose patients. This problem might be solved with more individualized sensor types, which are not available at the moment. Parallel measurement of BIS® and NOL® can be difficult, depending on the area that is operated on. Especially intracranial surgery is prone to losing BIS® electrode contact during the intervention or electrode placement might not be possible right from the start. However, combined measurement is not mandatory to perform NOL®-guided analgesia, yet it helps to differentiate between relevant nociceptive stimuli and mere lack of hypnosis.

Naturally, larger patient populations are necessary for a prospective design and adequate analysis, however, for the purpose of this study and as a primary evaluation, the patient numbers appear reasonable.

Interestingly, the aspect of hyperanalgesia is, in our opinion, not sufficiently addressed in current literature. It can be assumed, that the prevention of opioid overdoses might benefit postoperative outcome measures like delirium and pulmonary complications resulting from residual opioid effects. More comprehensive prospective study designs would be necessary to support these hypotheses.

Although all of our results look promising and align with other studies in different surgical contexts, nociceptive monitoring remains a challenge, since there is no direct way to measure nociceptive signalling. Hence, all available systems rely on surrogate parameters, usually depicting vegetative reactions of the sympathetic nervous system. Hence, irregularities, individual differences or direct anesthetic effects (e.g., sympathetic block due to peridural anesthesia) have to be accounted for. Sound clinical judgement remains paramount when making crucial decisions on patient treatment. However, nociception monitoring might be able to support those decisions in the future.

This preliminary pilot study was conducted to create the basis for the planning of more elaborate prospective trials. The focus on the nociceptive measurement technique in the perioperative phase results in several limitations regarding the assessment of broader clinical data. Firstly, the patient collective was completely unselected with the only inclusion criteria being intracranial tumor resection. There was no collection of oncologic data regarding pre- or postoperative therapy, malignancy or outcome measures and no longterm follow up, since no therapeutic intervention was performed and hence no inter-group effect would have been expected. Accordingly, all observed patients received the same standard of care during the surgery. Secondly, while there were no intraoperative complications in the reported patients, possible postoperative problems beyond the PACU stay were not analyzed and cannot be provided here. While the narrow focus of this first data set prevents any further clinical assessment, follow-up trials with a prospective design should include a more extensive collection of relevant clinical data.

Conclusion

Nociceptive monitoring with the NOL® system during intracranial surgical procedures is generally feasible and enables plausible assessment of nociceptive stimulation and adequate analgesia. Those results could be used to design prospective randomized trials in order to potentially reduce hyperanalgesia in those patients, possibly allowing for earlier emergence and better post-operative evaluation and treatment.

Supplemental Information

Supplemental Information 1 Raw patient data before statistical analysis.

Raw data sets of vital parameters (MAP, NOL, BIS etc.)

Click here for additional data file.

Supplemental Information 2 STROBE Checklist.

Consent was waived due to the retrospective nature of the study

Click here for additional data file.

This manuscript will be part of the doctoral thesis of Veselina Moravenova.

Abbreviation list

ANI® Analgesia Nociception Index

BIS® Bispectral index

BPM Beats per minute

HR Heart rate

MAP Mean arterial pressure

NOL® Nociception level monitoring

NPRS Numeric pain rating scale

PACU Post anesthesia care unit

SD Standard deviation

Additional Information and Declarations

Competing Interests

Author Contributions

Human Ethics

Clinical Trial Ethics

Data Availability

Clinical Trial Registration

Alexander Ziebart and Eva-Verena Griemert gave paid lectures for Medtronic. All other authors have no conflicts of interest to declare. The data included in this article have not been presented anywhere else.

Robert Ruemmler conceived and designed the experiments, performed the experiments, analyzed the data, prepared figures and/or tables, authored or reviewed drafts of the article, and approved the final draft.

Veselina Moravenova performed the experiments, analyzed the data, prepared figures and/or tables, and approved the final draft.

Sandy Al-Butmeh performed the experiments, prepared figures and/or tables, and approved the final draft.

Kimiko Fukui-Dunkel performed the experiments, prepared figures and/or tables, and approved the final draft.

Eva-Verena Griemert conceived and designed the experiments, analyzed the data, authored or reviewed drafts of the article, and approved the final draft.

Alexander Ziebart conceived and designed the experiments, analyzed the data, authored or reviewed drafts of the article, and approved the final draft.

The following information was supplied relating to ethical approvals (i.e., approving body and any reference numbers):

Ethics committee approval no. 2021-16201, regional ethics committee of Rhineland Palatine, Mainz, chairman Dr. Stephan Letzel. Due to regional regulations, the analysis of routine data does not have to be specifically approved and is exempt from individual consent.

The following information was supplied relating to ethical approvals (i.e., approving body and any reference numbers):

Ethics committee approval no. 2021-16201, regional ethics committee of Rhineland Palatine, Mainz, chairman Dr. Stephan Letzel.

The following information was supplied regarding data availability:

The measured data before statistical analysis is available in the Supplemental File.

The following information was supplied regarding Clinical Trial registration: DRKS00029120.

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
