# Peer review of "A novel non-invasive nociceptive monitoring approach fit for intracerebral surgery: a retrospective analysis"

_PeerJ, doi:10.7717/peerj.16787_

## Round 0.1 · original submission · Minor Revisions

Please do the revisions required.

Reviewer 1 has suggested that you cite specific references. You are welcome to add it/them if you believe they are relevant. However, you are not required to include these citations, and if you do not include them, this will not influence my decision.

Reviewer 1 ·

Basic reporting

To ensure patient’s safety, comfort, and success of operation, anesthesia depth should be measured effectively during intracerebral surgery, especially when the head is immobilized in a Mayfield-clamp. Electro-encephalography measurements are an established method to measure deep sedation. One major caveat, though, is the lack of nociception visualizing. In this single-center, retrospective observation study, the authors collected nociceptive data via a nociception level monitor of 40 patients (age 56 ± 18 years) undergoing intracerebral tumor resection and investigated whether this monitoring is feasible and delivers relevant values to potentially base therapeutic decisions on. In their setup, the patients received total intravenous anesthesia and were connected to the device via a finger clip as well as bispectral-index monitoring to confirm deep sedation. The results indicated that the largest increase in nociceptive stimulation occurred during intubation, Mayfield positioning and incision. Therefore, it can be concluded that nociceptive monitoring using the NOL system during intracerebral surgery is feasible and can result in yielding helpful information to support therapeutic decisions.

This is an interesting study which can add to the body of our knowledge on an important topic, which can have an immediate and direct effect on patients’ outcomes. However, I have some suggestions for the authors which need to be addressed before the manuscript is considered further. I provide detailed comments below.

Experimental design

Methods: Line 110: “haemodynamics” is British spelling, whereas “anesthesia” that is used in the manuscript is the American spelling, and sometimes “anesthaesia” which is British spelling is used. It does not matter which spelling is used, but the manuscript should adopt one form and be consistent.

Methods: What was the inclusion/exclusion criteria? How the patients were selected?

Validity of the findings

Results: More information on patients should be provided: what was the age range in addition to the mean and SD?

Results: What was the tumour type?

Results: How long did the surgery range in mean and range?

Results: Were there any intraoperative complications in surgery?

Results: Did patients receive other therapies (chemo/radio) in addition to surgery?

Results: Was there any association between NOL and tumor type/surgery duration/age/etc?

Overall, Results section needs expansion.

Discussion is well-written.

Discussion: Line 172: Change “allow for” to “enable” to improve readability

Discussion: “Sound clinical judgement remains paramount when making crucial decisions on patient treatment.” This is a suggestion- it can be considered to discuss the role of artificial intelligence in taking into account all physiological and pathological variables in different patients to make the most optimal decision. The following paper can be cited discussing the role of AI: Neurosurgery and artificial intelligence. AIMS Neurosci. 2021 Aug 6;8(4):477-495. doi: 10.3934/Neuroscience.2021025. PMID: 34877400;

The limitations of the study should be discussed more clearly, such as small sample size, retrospective design, etc in addition to technical limitations.

Additional comments

Abstract: Background: Please mention the aim of the study in one sentence as the last sentence of the abstract background rather than in Methods.

Abstract: Line 31: Please delete “retrospectively” as it has been mentioned previously in line 29.

Background: Line 74: Change “up to” to “including”

Background: Line 77: Change “on the other hand” to “in addition”

Background: Please mention some statistics on anesthesia failure during surgery as there are many papers on intraoperative and anesthesia awareness

Background: Line 79: This sentence needs a reference: “However, hyperanalgesia with opioid overdosing or unnecessarily deep hypnosis and muscle relaxation can be detrimental to that goal.”

Background: Line 87: Change “one of those devices” to “a such device”

Background: Line 87: These sentences need references: “The PMD-200 Nociception Level monitor (NOLÆ) is one of those devices, established by attaching a sensor to the patientís finger. The multi-channel assessment simultaneously measures temperature, perfusion, movement and skin conductivity and processes those measurements to a dimensionless score reaching from 0 (no activity) to 100 (maximum activity/pain) with a threshold of sufficient analgesia from 10-35.”

·

Basic reporting

This is a noteworthy retrospective observational study conducted at a single center, focusing on precisely quantifying a patient's physiological pain response during intracerebral tumor resection using a NOL® (Nociception Level) monitor. The study aims to assess the viability of an NOL-guided analgesia protocol to inform therapeutic decisions, particularly in predicting postoperative pain and optimizing opioid use based on NOL® nociceptive monitoring data.

The findings of this study have the potential to mitigate hyperanalgesia, shorten emergence periods, and reduce postoperative complications. They collected and analyzed 40 intravenous anesthesia patients’ NOL and BIS data, evaluated four key time points of nociceptive stress (intubation, Mayfield-positioning, incision, extubation), and compared them to standard vital signs.

They found the nociceptive stimulation in descending order were intubation, Mayfield-positioning, and extubation.

Experimental design

1. Currently, there is no direct method for measuring nociceptive levels. The NOL® monitor relies on surrogate parameters derived from the autonomic nervous system's vegetative reactions. To enhance the monitoring of nociceptive levels at four critical time points, could we integrate Analgesia/Nociception Index (ANI) data, which assesses heart rate variability and parasympathetic tone? Using both NOL® and ANI monitors, this combined approach could yield more dependable data to inform therapeutic decisions.

2. Is it feasible to increase the sample size?

3. Establish rigorous admission criteria for observational data to eliminate potential confounding factors, such as arrhythmias or patients currently undergoing beta-blocker therapy.

Validity of the findings

In Figure 3, when we examine the nociception levels at the extubation time point, we observe an approximately twofold increase compared to the baseline measurements. This increase implies that the analgesia provided may be insufficient for effectively managing the nociceptive stimuli during extubation (with a target NOL® goal of <25). Could you please offer some guidance on adjusting the analgesia dosage at this moment, considering the established baseline conditions?

Reviewer 3 ·

Basic reporting

• Original primary research within aims and scope of the journal.
• Research question well defined, relevant & meaningful. It is stated how research fills an identified knowledge gap.
• Methods described with sufficient detail & information to replicate.

Experimental design

• Original primary research within aims and scope of the journal.
• Research question well defined, relevant & meaningful. It is stated how research fills an identified knowledge gap.
• Methods described with sufficient detail & information to replicate.

Validity of the findings

The findings were valid and rationale to the objective of the study. However, it was more on descriptive results.

• Conclusions are well stated, linked to original research question & limited to supporting results.

Additional comments

• This was a pilot study and the results were more on descriptive. However, it gave some useful information for the future study on the usage of NOL as a tool of nociceptive monitoring during neurosurgery.

---

## Round 0.2 · Major Revisions

Dear Author, You will need to revise seriously your manuscript to the suggestions that were given by the peer reviewer if not it has to be rejected.

Reviewer 1 ·

Basic reporting

There are multiple comments which have not been addressed, and I am not convinced with authors reply.

Experimental design

Please ensure the following questions are addressed, which are important from a neurosurgical point of view:

Results: More information on patients should be provided: what was the age range in addition to the mean and SD?

Results: What was the tumour type?

Results: How long did the surgery range in mean and range?

Results: Were there any intraoperative complications in surgery?

Results: Did patients receive other therapies (chemo/radio) in addition to surgery?

Results: Was there any association between NOL and tumor type/surgery duration/age/etc?

Validity of the findings

More information is required to validate the findings, and once the previous questions are addressed, the findings can be evaluated.

---

## Round 0.3 · Major Revisions

Please add in all limitation statements. Thank you

Reviewer 1 ·

Basic reporting

I thank the authors for their response. While in the response letter the authors have explained and acknowledged the limitations of the study, such caveats and shortcomings schould be mentioned in the manuscript to avoid misleading the readers. Therefore, a limitation paragraph is required to address the following problems and provide solutions in future studies:

1. "No perioperative complications occured for the surgeries analyzed in this study. However, since no follow-up period was part of this study, the authors are not able to provide any postoperative data outside of the PACU."

2. "Since this was an unselected patient collective without any focus on treated/untreated intracranial tumours, surgical and oncological data were not collected for the study population and cannot be provided here. "

3- "All patients belonged to an unselected population with the only criterium being “intracranial tumor resection”. Accordingly, no specific association with singular tumor entities was seen and all NOL patients received the same standard of care."

Experimental design

There are major concerns about the experimental design and some important factors have been ignored. It is understandable that the focus of the study can be different, however, this should not prevent data collection.

Validity of the findings

The findings are limited and many factors are not included. Therefore, such limitations can influence the outcome. Discussing those caveats can raise the awareness among the audience. Despite this, there are serious flaws in the study design and data collection.

---

## Round 0.4 · accepted · Accept

With all limitations explained in the text, this manuscript is accepted.